# Recent Advances on the Anti-Inflammatory and Antioxidant Properties of Red Grape Polyphenols: In Vitro and In Vivo Studies

**DOI:** 10.3390/antiox9010035

**Published:** 2019-12-31

**Authors:** Thea Magrone, Manrico Magrone, Matteo Antonio Russo, Emilio Jirillo

**Affiliations:** 1Department of Basic Medical Sciences, Neuroscience and Sensory Organs, School of Medicine, University of Bari, 70124 Bari, Italy; manrico.magrone@gmail.com (M.M.); emilio.jirillo@uniba.it (E.J.); 2MEBIC Consortium, San Raffaele Open University of Rome and IRCCS San Raffaele Pisana of Rome, 00166 Rome, Italy; matteo.russo@sanraffaele.it

**Keywords:** red grape polyphenols, immunity, inflammation, obesity, allergy, cancer, cellular and molecular rehabilitation

## Abstract

In this review, special emphasis will be placed on red grape polyphenols for their antioxidant and anti-inflammatory activities. Therefore, their capacity to inhibit major pathways responsible for activation of oxidative systems and expression and release of proinflammatory cytokines and chemokines will be discussed. Furthermore, regulation of immune cells by polyphenols will be illustrated with special reference to the activation of T regulatory cells which support a tolerogenic pathway at intestinal level. Additionally, the effects of red grape polyphenols will be analyzed in obesity, as a low-grade systemic inflammation. Also, possible modifications of inflammatory bowel disease biomarkers and clinical course have been studied upon polyphenol administration, either in animal models or in clinical trials. Moreover, the ability of polyphenols to cross the blood–brain barrier has been exploited to investigate their neuroprotective properties. In cancer, polyphenols seem to exert several beneficial effects, even if conflicting data are reported about their influence on T regulatory cells. Finally, the effects of polyphenols have been evaluated in experimental models of allergy and autoimmune diseases. Conclusively, red grape polyphenols are endowed with a great antioxidant and anti-inflammatory potential but some issues, such as polyphenol bioavailability, activity of metabolites, and interaction with microbiota, deserve deeper studies.

## 1. Introduction

Polyphenols are phenolic compounds largely spread in the vegetal kingdom where they play a protective role coping with several environmental insults (e.g., ultraviolet lights, free radicals, and temperatures) [1,2,3]. For instance, in the Mediterranean area, olives and grapes have been demonstrated to increase polyphenol production due to their high sensitivity to stressors [4]. In nature, more than 8000 different polyphenols exist as major components of fruits, vegetables, cereals and their derivatives (wine, extra virgin olive oil, chocolate, and juices) [1,2,3], and structurally can be divided into, flavonoids and non-flavonoids compounds.

Flavonoids are based on a common structure composed by two aromatic rings which are bound by three carbon atoms, finally, forming an oxygenated heterocycle [5]. On the other hand, stilbenes and, especially resveratrol (RES), represent the non-flavonoid components present in low amounts in human diet [1,2,6,7]. They are composed by two phenyl rings bound together by two carbon methyl bridges [1,2].

In this framework, it is worthwhile mentioning some polyphenols present in extra virgin olive oil for their antioxidant and anti-inflammatory properties. For instance, lignans are fiber-associated polyphenols whose structure is based on a 2,3-dibenzylbutane complex, derived from the dimerization of two cinnamil acid residues [8]. Finally, thyrosol-derived compounds, such as oleuropein and hydroxytyrosol, are the main polyphenols in extra virgin olive oil [9,10,11]. Chemically, thyrosols are represented by a phenethyl alcohol moiety with a hydroxyl group at the fourth position of the benzene group.

Polyphenol activity depends on their absorption rate and bioavailability of derivative metabolites. In particular, once ingested, polyphenols interact with other nutrients such as proteins, sugars, fats, fibers and the intestinal microbiota, thus leading to the generation of active metabolites [12]. Polyphenol absorption is a quite complex process since the majority of them are present as glycosides, i.e., conjugated with sugars. Specifically, anthocyanins are absorbed intact, while others are converted into aglycones via hydrolysis by the small intestine brush border (via hydrolase) or within epithelial cells (via cytosolic β-glucosidase or lactase phlorizin) in the colon [13,14,15]. In turn, aglycones pass to the circulation under conjugated forms, such as sulfate, glucuronide, and/or methylated metabolites, this occurring within epithelial cells and in the liver [15]. Finally, aglycones undergo ring fixation with production of bioactive metabolites, such as phenolic acids and hydroxycinnamates, which can be detected in the plasma after 12–48 h from polyphenol ingestion.

Dietary polyphenols and fruit-derived polyphenol supplements contain a large array of different polyphenols and, therefore, the mechanism of ingestion and metabolite production are more complex, also depending on individual variations of microbiota composition [16]. Human beings acquire polyphenols trough diet as in the case of Mediterranean-type diet (Med) [17,18]. In particular, dietary flavonoids are the most common polyphenols which exert healthy effects in terms of metabolism, weight, chronic disease, and neuroendocrine immune control [19,20,21].

Here, emphasis will be placed on red grape polyphenols. For instance, wine polyphenols represent an important dietary source with flavonoids accounting for >85%, ≥1 g/L of total phenolics [22]. A minor component is represented by derivatives of carboxylic acids, hydroxycinnamate, tannins, and RES [23]. Flavonoids are extracted from grape skin, seeds, and stem, whereas tannins are present in oak barrels during wine storage. RES is present in the grape as a result of several insults, such as mechanical trauma, infections with fungi, and ultraviolet light radiations [24]. The healthy properties of red wine have been emphasized in the context of the French paradox since in France (e.g., Bordeaux region) the low incidence of cardiovascular disease has been attributed to the moderate consumption of red wine in comparison to other western countries [25,26,27]. However, other authors have confuted the French paradox claiming that reported healthy effects originate from MeD adoption and not only from red wine intake [28,29,30].

Aim of the present review will be to describe and discuss the effects of red grape polyphenols in experimental and clinical settings with special reference to their antioxidant and anti-inflammatory properties.

## 2. Antioxidant and Anti-Inflammatory Activities Exerted by Red Grape Polyphenols

There is a wealth of information on the ability of dietary polyphenols to exert antioxidant functions, scavenging reactive oxygen species (ROS), as well as anti-inflammatory activities, altering the expression of genes like proinflammatory cytokines, lipoxygenase (LOX), nitric oxide synthase (NOS), and cyclo-oxygenase (COX) [31,32,33,34,35,36]. ROS production is associated with oxidative stress and protein oxidation which, in turn, account for induction of the inflammatory pathway [37,38]. Therefore, interruption of the oxidative process (e.g., ROS generation) attenuates triggering of the inflammatory cascade. Polyphenols have been shown to exert antioxidant activity scavenging radicals and chelate metal ions (e.g., quercetin chelates iron ion) [39]. Polyphenol-induced metal ion chelation reduces the formation of O_2_^•^ in Chlamydia-primed THP1-monocytes, also protecting endothelial cells from oxidative insults [40,41]. Other antioxidant mechanisms elicited by polyphenols are represented by blockade of the mitochondrial respiratory chain and adenosine triphosphatase and xantine oxidase [42,43,44]. Finally, curcumin and epigallocatechin gallate (EGCG) are able to activate antioxidant enzymes, such as superoxide dismutase, catalase, and glutathione peroxidase, thus leading to ROS detoxification [45,46].

With special reference to red grape polyphenols, RES could inhibit COX, peroxisome proliferator activated receptor-γ and endothelial NOS in vitro and in vivo experiments with murine and rat macrophages [47,48,49]. In this context, polyphenols extracted from high EGCG content Canosina red grape cultivar were able to inhibit either in vitro or in vivo release of nitric oxide (NO) from human monocytes of patients with nickel (Ni)-mediated contact allergic dermatitis (CAD) [50,51,52].

### 2.1. Regulation of NF-κB

Quercetin and EGCG—other flavonoids present in red grapes—are able to inactivate nuclear factor kappa-light-chain-enhancer of activated B cells (NF-κB) in human epithelial cells and human monocytes [53,54], thus leading to inhibition of proinflammatory cytokines, chemokines, adhesion molecules, and growth factor release [55]. Particularly, by using quercetin the molecular mechanisms implicated in deactivation of NF-κB nuclear translocation have been elucidated. This flavonoid, prevented the nuclear translocation of p50 and p65 subunits of NF-κB, as well as the phosphorylation of IκB kinase (IκB)α proteins in macrophages [56,57]. Also, in human mast cells, quercetin blocked the activation of NF-κB through the above cited mechanisms, thus, decreasing release of tumor necrosis factor (TNF)-α, interleukin (IL)-1β, IL-6 and IL-8 [58]. In mouse BV-2 microglia treated by lipopolysaccharides (LPS) and interferon (IFN)-γ, quercetin hampered the binding of NF-κB to DNA, thus preventing release of proinflammatory cytokines [59]. In sum, flavonoids are able to regulate NF-κB activation either at early phases, inhibiting Iκκ activation or at late stages, preventing binding of NF-κB to DNA [60,61,62].

### 2.2. Regulation of Mitogen-Activated Protein Kinases

The mitogen-activated protein kinases (MAPKs) regulate gene transcription and transcription factor activities implicated in inflammation. Among them, extracellular signal-related kinases (ERKs)-1, -2, c-Jun amino-terminal kinases (JNK)-1/2/3, p-38-MAPKs, and ERK-5 are able to interact with NF-κB, thus, suggesting the intricacy of MAPK pathway. Evidence has been provided that both quercetin and EGCG interfere with the MAPK signaling system reducing production of TNF-α and IL-12 in immune and non-immune cells [63,64]. The above cited anti-inflammatory mechanisms mediated by catechin and quercetin have also been reported to occur in mouse skin [65], and in human coronary endothelial cells [66]; thus, indicating the protective role of these compounds in inflammation.

### 2.3. Regulation of Arachidonic Acid

Among other mechanisms of anti-inflammation promoted by polyphenols, inhibition of arachidonic acid (AA) pathway plays a paramount role. AA is released by membrane phospholipids following phospholipase A (PLA)2 cleavage. In turn, AA is metabolized by COX and LOX with generation of prostaglandins (PGs) and thromboxane A2 by COX and leukotrienes (LTs) by LOX [67]. Polyphenols are able to reduce release of PGs and LTs via inhibition of PLA2, COX, and LOX, as experimentally seen with quercetin, red wine, and EGCG [68,69,70]. Quite interestingly, some polyphenols share structural and functional similarities with anti-inflammatory drugs as in the case of oleocanthal, which mimics the activity of ibuprofen, inhibiting COX-1 and COX-2 [71].

For the sake of clarity, evidence has been provided that LOX may act as a pro-resolving mediator in the resolution on neo-intimal hyperplasia [72]. Also, PGE2 has been shown to play an anti-inflammatory role in allergen-induced airway response when inhaled by asthma patients [73].

Major antioxidant and anti-inflammatory effects exerted by red wine polyphenols are illustrated in Table 1.

## 3. Regulation of Immune Functions by Polyphenols

### 3.1. Receptors for Polyphenols

There is a large body of evidence that polyphenols can regulate immune functions via binding to various receptors. Aryl hydrocarbon receptor (AhR) is located on the cytoplasm of several immune and non-immune cells in association with heat shock protein 90 and the co-chaperone 23 [74]. At intestinal level, AhR has been found in the cytoplasm of intraepithelial lymphocytes, innate lymphoid cells, dendritic cells (DCs), macrophages and T helper (h)-17 cells. Then, dietary polyphenols binding to AhR may modulate gut immune response. For instance, dietary naringenin induces T regulatory (Treg) cells binding to intestinal AhR [75]. Furthermore, EGCG is able to bind to the 67 kDa laminin receptor, the zeta-chain-associated 70kDa protein (ZAP-70), and the retinoic acid-inducible gene (RIG)-I, respectively [75,76,77]. Neutrophils, monocytes/macrophages, mast cells, and T cells express ZAP-70 [78,79]. Inhibition of ZAP-70 by EGCG regulates CD3-mediated T cell receptor signaling in leukemic cells [80]. EGCG also suppresses signaling by the dsRNA innate immune receptor RIG-I [81]. Specific protein 1 is a transcription factor expressed on many cancer cells and its inhibition by RES suppresses growth of human mesothelioma cells [82]. Other receptors, such as Toll-like receptor (TLR)-4, T cell receptor-αβ and surface IgM B cell receptor are common binding sites for baicalin, a flavone glycoside [83], thus leading to innate and adaptive immune response modulation.

### 3.2. Anti-Inflammatory Mechanisms

As reported by in vitro and in vivo studies, polyphenols contained in red grapes and red wines are able to perform a potent immunomodulation. Quercetin treatment of DCs led to reduced production of proinflammatory cytokines and chemokines with a decrease in Major Histocompatibility Complex class II and costimulatory molecules in the context of the immunological synapsis [84]. Consequentially, evidence has been provided that quercetin-induced deactivation of LPS-stimulated DCs down-regulates T cell response to specific antigens [85]. Similar results have been obtained in vitro treating peripheral human monocytes from healthy donors with red wine-derived polyphenols, even including quercetin [86]. Particularly, co-incubation of monocytes with polyphenols and LPS abrogated the LPS-mediated activation of NF-κB likely by a phenomenon of steric hindrance. As a result of such an inhibitory mechanism, the storm of proinflammatory cytokines released by human monocytes was noticeably attenuated [87]. In the same direction, in vitro quercetin treatment of peripheral blood mononuclear cells from multiple sclerosis patients reduced release of IL-1β and TNF-α, and this effect was potentiated in the presence of IFN-β [88].

Fisetin is a flavonoid contained in a number of plants and fruits, even including grapes. Fisetin has been shown to in vitro inhibit production of Th1 and Th2-related cytokines and modify the ratio CD4+/CD8+ T cells [89].

This effect seems to depend on the down-regulation of NF-κB activation and nuclear factor of activated T cell signaling. In vivo, fisetin suppressed murine delayed-type hypersensitivity reactions, thus supporting its inhibitory role on T cells [89]. RES exerts anti-inflammatory and immunomodulating functions through activation of sirtuin-1 (Sirt-1) [90]. Sirt-1 operates by disrupting the TLR-4/NF-κB/signal transducer and activator of transcription (STAT) pathway with decreased production of cytokines, platelet activating factor and histamine [91,92]. Sirt-1, as a deacetylase, plays an important role in immune tolerance and its abrogation leads to a spontaneous development of autoimmune disease [93,94]. RES binding to Sirt-1 enhances its attachment to p65/RelA substrate [95], which, as a member of the NF-κB pathway, activates leukocytes and the proinflammatory cytokine pathway [96]. Then, Sirt-1 activation by RES hampers RelA acetylation with decrease of NF-κB-induced expression of TNF-α, IL-1β, IL-6, metalloproteases (MMPs), and COX-2 [93]. As recently reviewed by Malaguarnera [97], RES induces AMP-activated protein kinase which, in turn, controls Sirt-1 activity, regulating the cellular levels of nicotinamide adenine dinucleotide (NAD+). The, NAD+-induced Sirt-1 activation leads to deacetylation and activation of peroxisome proliferator-activated receptor γ coactivator-1α.

Quite importantly, the anti-inflammatory activity mediated by RES via activation of Sirt-1 is abrogated by genetic deletion of Sirt-1 or its inhibitors such as sirtinol [98,99,100]. Furthermore, RES is able to modulate macrophage function acting upon TLR-4 and TRAF5-mediated inflammatory responses, deactivating LPS-dependent NF-κB activation and COX-2 expression [101,102].

Nucleotide oligomerization domain-like receptors (NLRs) belong to the pattern recognition receptor family and their activation is involved in the development of inflammatory diseases. In this respect, evidence has been provided that RES inhibits the increase of α-tubulin-mediated assembly of the NLR pyrin domain containing 3 (NLRP)3 inflammasome [103]. Therefore, RES may represent an important therapeutic tool in the management of NLRP3-inflammasome-induced disease.

### 3.3. Modulation of Cytokines Production

Several reports have demonstrated the ability of RES to modulate cytokine production, e.g., inhibiting release of granulocyte-macrophage colony-stimulating factor, IL-1β, and IL-6; thus, attenuating low grade chronic inflammation as well as atheroma formation [104,105,106,107].

With special reference to T cells, RES exerts anti-inflammatory effects, reducing numbers of Th17 cells and production of IL-17, an inflammatory cytokine, in murine collagen-induced arthritis [108]. On the other hand, it is well known that RES mediates T cell tolerance via upregulation of Sirt-1 in activated T cells [109]. In the same direction, another report has demonstrated that RES increased release of IL-10, an anti-inflammatory cytokine produced by Treg cells [110]. Similar results were attained stimulating human healthy peripheral blood lymphocytes with polyphenols from fermented grape marc (FGM), thus, leading to induction of FoxP3+ Treg cells and enhanced release of IL-10 [111]. However, other data have reported a RES-mediated suppression of CD4+CD25+ cells with decreased production of transforming growth factor (TGF)-β and enhanced expression of IFN-γ in CD8+ cells [112].

With special reference to natural killer (NK) cells, RES has been shown to enhance their killing activity against leukemia and lymphoma cells [113]. In another study, evidence has been provided on the capacity of RES to up-regulate perforin expression on NK cells; thus, supporting the enhancement of their lytic activity [114]. Also, in an infectious model of acute pneumonia in rats, RES treatment increased NK cell activity which correlated with a decreased bacterial burden and mortality [115].

Polyphenol-mediated immunomodulation is described in Table 2.

## 4. Polyphenol-Mediated Immune Responses in Pathological Conditions

In this review, the illustration of antioxidant and anti-inflammatory effects exerted by polyphenols will be restricted to major pathologies such as obesity, inflammatory bowel disease (IBD), cancer, neurodegeneration, and allergy/autoimmunity.

### 4.1. Obesity

Overweight/obesity is pandemic and affects more than 2.5 billion adults, even including those living in developing countries [116,117]. Of importance, obesity leads to the outcome of metabolic syndrome, such as type 2 diabetes, cardiovascular disease, neurodegeneration, and cancer [118]. Obesity can be defined as a low grade chronic inflammation maintained by the visceral adipose tissue, as a continuous source of inflammatory mediators [119,120]. In particular, obesity is characterized by an exaggerate lipolysis with secretion of free fatty acids, which, in turn, trigger inflammatory responses, production of ROS, and insulin resistance [121,122]. On these grounds, a number of experimental and clinical studies have been focused on the effectiveness of polyphenols to attenuate the oxidative/inflammatory status in obesity. Gallic acid, as a component of red grape polyphenols, is able to decrease body weight in obese rodents, inhibiting lipid droplet formation in the liver or adipose tissue, as well as reducing serum levels of triglycerides and low density lipoproteins and improving glucose tolerance [123,124,125,126]. There is evidence that gallic acid controls glucose and lipid metabolism, regulating phosphatidylinositol 3-kinase (PI3K)/protein kinase B (AKT) and AMPK signaling pathways [127]. In obese people, clinical trials based on the administration of gallic acid have been quite controversial. Two studies failed to demonstrate weight loss or reduction of markers associated to obesity upon administration of gallic acid, as reported by [121]. On the other hand, other investigations documented that administration of gallic acid reduced waist circumference, body mass index (BMI), and visceral fat in pre-obese individuals, also decreasing oxidative and inflammatory markers [128,129,130,131]. It is likely that divergent results obtained with gallic acid may depend on patient selection since more efficacy has been observed in those trials with pre-obese people.

With special reference to peripheral immune markers, red grape polyphenols extracted from Nero di Troia cultivar were in vitro used to stimulate blood lymphomonocytes isolated from obese people. This treatment was able to reduce the inflammatory status of obese lymphomonocytes, decreasing release of IL-17 and IL-21 (an inducer of Th17 cells), while enhancing production of IL-10 [132]. At the same time, release of IL-1β and TNF-α also dramatically dropped.

These data indicate the imbalance of peripheral immune responses in obese people and the ability of polyphenols to attenuate inflammatory biomarkers.

There is evidence that childhood obesity is increasing, thus representing an emerging clinical problem worldwide [133]. In this respect, unhealthy dietary habits predispose to childhood obesity, as reported in a group of normal weight children under a MeD regimen for one year [134]. In fact, those children, who disattended dietary advice, increased BMI, salivary levels of IL-17, and decreased salivary IL-10 amounts. Conversely, in children who attended MeD IL-10 levels increased with a reduction of IL-17 salivary levels.

These results indicate that MeD, based on polyphenols, unsaturated fatty acids, vitamins and oligoelements can prevent overweight/obesity in early childhood [134].

Diabetes is very often associated to obesity and evidence has been provided that polyphenols (e.g., quercetin and epicatechins) can also correct diabetic complications [135,136,137,138]. In particular, experiments with insulin releasing cell lines and isolated pancreatic islets have demonstrated that polyphenols protect β cell survival, inhibiting NF-κB activation, triggering the PI3K/AKT pathway while inhibiting ROS generation [139].

Even if lack of clinical trials on the effects of flavonoids on β cells represents a limitation of the above reported experimented data, nevertheless, flavonoids have been shown to exert anti-hyperglycemic activity in diabetic patient [140,141]. According to Ghorbani [139] the anti-hyperglycemic effects mediated by flavonoids may be ascribed to decrease in glucose absorption, improved insulin resistance, enhanced insulin secretion from β cells, and inhibition of gluconeogenesis.

Major effects of polyphenols on obesity/diabetes are expressed in Table 3.

### 4.2. Inflammatory Bowel Disease

IBD are chronic pathologies of the intestinal mucosa exhibiting a multiple pathogenesis. In fact, genetic factors, abnormal functions of the immune response, alteration of the intestinal barrier and dysbiosis seem to contribute to disease outcome and maintenance [142,143,144,145].

The beneficial effects of polyphenols have been evaluated in the course of experimental colitis [146,147]. Red grape polyphenols extracted from FGM were able to attenuate dextran sulfate sodium (DSS) murine colitis when orally administered [148]. This experimental regimen abrogated shortening of intestine length and reduced content of IL-1β and TNF-α in intestinal homogenates from treated mice. In a recent paper, administration of bronze tomatoes, enriched in flavonols, anthocyanins and stilbenoids, as well as red grape skin, reduced intestinal damage in the course of DSS-induced experimental colitis with improvement of stool consistency, fecal blood content, and weight loss [149].

In two rat model of 2,4,6-trinitrobenzenesulfonic acid, RES mitigated intestinal inflammation decreasing PG production, COX-2 expression, neutrophil recruitment and TNF-α secretion [150], also regulating genes involved in IL-6 signaling, apoptosis, mitochondria fatty acid oxidation, and Wnt-signaling [151]. In a model of DSS-induced murine colitis, oral administration of RES was effective in the inhibition of inducible NOS expression and NF-κB activation, thus, preventing the onset of intestinal inflammation [152].

The IL-10^−/−^ mouse model represents a suitable model of IBD [153]. In these mice, administration of RES induced activation of myeloid-derived suppressor cells (MDSCs), thus attenuating mucosal and systemic inflammation [154].

As recently reviewed by Nunes and associates [155], RES administration to mice with DSS-induced ulcerative colitis (UC) decreased inflammatory and oxidative markers, also ameliorating clinical symptoms (loss of body weight, diarrhea, and rectal bleeding) [156], and reducing rate of mortality [157]. In another study dealing with a DSS-induced murine model of UC, RES was able to modulate Th17/Treg cell ratio, decreasing number of the former and upregulating number of the latter [158].

With special reference to clinical trials, Samsami-Kor and associates [159] evaluated the effects of RES supplementation (0.5 g/day for 6 weeks) in a group of patients affected by UC. C-reactive protein (C-rp), TNF-α, and NF-κB levels decreased with an improvement of clinical colitis activity index score. Finally, in RES-treated patients superoxide dismutase and total antioxidant capacity increased, while malondialdehyde levels decreased.

In Table 4 effects of polyphenols on IBD are illustrated.

### 4.3. Neurodegeneration

Among neurodegenerative disorders, Alzheimer’s disease (AD) and Parkinson’s disease (PD) are increasing also in relation to life style changes, aging, environmental, and genetic risk factors. Quite interestingly, polyphenols have been experimented in vitro and in vivo models of AD and PD, in view of their ability to cross the blood brain barrier (BBB) and accumulate into the brain. For instance, in an in vitro model, penetration of methylated conjugates of polyphenols through the BBB was higher than that of sulfated or glucuronidated molecules [160,161]. Another report demonstrated catechin and epicatechin transport across BBB [162].

In vivo studies have shown the ability of RES, EGCG, quercetin, cathechins and curcumin to accumulate into the central nervous system [163,164,165,166,167]. There is also evidence that persistent intra-gastric administration of EGCG led to an elevated concentration of the aglycone form (5–10% of plasma concentrations) in various organs, even including brain [164].

Another important aspect of the neuroprotective effects of polyphenols is their capacity to act synergistically. Combinations of RES and catechins exhibited a synergistic protective activity against amyloid (A)β toxicity, oxidative stress, and oxygen-glucose deprivation in vitro [168,169,170,171]. Synergy has also been shown between polyphenols, drugs, and hormones. For instance, a potentiation of effects on neurite outgrowth has been reported, in vitro using the combination brain-derived neurotrophic factor and catechins [172]. In a murine model of PD, rasagiline, an inhibitor of dopamine metabolizing monoamine oxidase B, synergized with polyphenols in promoting survival of the dopaminergic nigrostriatal pathway [173,174,175]. In this context, a *Vitis vinifera* red grape seed and skin extract (GSSE) exhibited in vitro and in vivo neuroprotective activity in a mouse model of PD [176]. GSSE protected dopamine neurons from neurotoxin 6-hydroxydopamine (6-OHDA) damage, reducing apoptosis, ROS production, and inflammatory markers. Also, motor function was improved in the same model of 6-OHDA-induced PD.

As recently reviewed by Azam and associates [177], TLRs are involved in the pathogenesis of neurodegenerative disorders. For instance, quercetin loaded into nanoparticles prevented AD progression via inhibition of TLR-4 signaling [178]. In addition, it decreased expression of TLR-4 and TLR-2, thus hampering proinflammatory cytokine production [179]. RES was shown to attenuate LPS and Aβ-mediated microglia neuroinflammation, inhibiting the TLR-4/NF-κB/STAT pathway [180]. EGCG was able to abrogate LPS-impaired adult hippocampal neurogenesis, silencing the TLR-4 signaling in mice [181,182,183].

Until now, a few clinical trials have been conducted to evaluate the efficacy of polyphenols in human neurodegeneration. RES administration has been found to attenuate neuroinflammation, cognitive decline and reduce liquoral levels of Aβ40 in AD patients [184,185]. Prolonged administration of RES and cocoa flavonols increased dentate gyrus-related cognitive functions and hippocampal memory [186,187,188].

The PROMESA-protocol is a phase III clinical testing based on daily oral treatment of 400 mg EGCG for 48 weeks in multiple system atrophy (MSA) patients [189]. MSA is a rare neurodegenerative disease where aggregation of α-synuclein in oligodendrocytes and neurons has been found. The above-indicated treatment did not modify disease progression in MSA and hepatotoxicity was reported in a few cases [190].

In Table 5, effects of polyphenols on neurodegeneration are described.

### 4.4. Cancer

Immune escape mechanisms evoked by cancer cells have extensively been explored and readers are referred to pertinent reviews for further details [191,192,193]. Particularly, immune suppression in cancer is mediated by Treg cells, MDSCs, and tumor-associated macrophages (TAMs) [191,194,195]. Here, the effects of polyphenols on these suppressive cells in cancer will be described.

With special reference to Treg cells, RES administration could decrease their frequency in mice bearing renal carcinoma [196]. In a model of Eg7 (syngenic lymphoma)-bearing C57BL/6 mice RES treatment led to a dramatic reduction of Treg cell percentage and TGF-β production, whereas intranodal CD8+ cells increased release of IFN-γ [197].

In a clinical trial based on the oral administration of EGCG for 6 months to chronic lymphocytic leukemia patients Rai stage O, a sharp decrease of Treg cells and of IL-10 and TFG-β in serum was detected [198]. Of note, despite the above cited examples of Treg cell suppression by polyphenols, other reports failed to demonstrate clear-cut effects of polyphenols on Treg cells [199,200].

As far as TAMs are concerned, these cells resemble M2 macrophages which promote tumor progression [201]. Strong evidence has been provided on the ability of RES to inhibit TAM activation via suppression of STAT3. This has been demonstrated in a lung cancer xenograft model where RES inhibited proliferation and expression of p-STAT-3 [202]. In another study, RES inhibited lymphangiogenesis in the context of a tumor, suppressing differentiation and activation of M2 macrophages [203]. The effects of polyphenols on MDSCs have also been demonstrated with other polyphenols such as curcumin. In mice bearing 4NQO-induced oral squamous carcinoma and in mice challenged with B16F10 melanoma cells lines, curcumin administration led to a dramatic reduction of MDSCs [204,205]. In a large-cell carcinoma lung cancer model, administration of curcumin reduced MDSCs in spleen and tumor infiltrates, increasing frequency of CD4+ and CD8+ cells, while decreasing IL-6 levels [206].

Other few studies have been focused on the effects of red wine extract (RWE) on cancer cell progression [207]. In BALC/c mice, RWE reduced growth of C26 cancer, suppressing angiogenesis and promoting apoptosis [208]. In preclinical studies, mice administered with RWE underwent a dramatic reduction of precancerous lesions in the colon [209,210]. In particular, reduction of fecal excretion of nitrosyl iron seems to play a fundamental role in the above model of inhibition of precancerous lesions [210]. Furthermore, evidence has been provided that muscadine grape skin extract was able to induce an unfolded protein response-mediated autophagy with apoptosis of human prostate cancer cells [211]. In this framework, Liofenol™ a RWE enriched in polyphenols, reduced colon cancer cell growth with an increase in p53 and p21 protein expression [212].

Polyphenol effects on cancer are summarized in Table 6.

### 4.5. Allergy and Autoimmune Diseases

Nowadays, allergic and autoimmune diseases are increasing; thus, likely depending on environmental factors and/or modifications of skin, lung and intestinal microbiota [213].

Polyphenol effects have been evaluated in various allergic and autoimmune conditions [214].

In vitro studies conducted with FGM from red grapes have demonstrated their ability to inhibit IgE binding to rat basophilic leukemia cells and to reduce human basophil degranulation [215,216]. Polyphenols extracted from seeds of red grape (Nero di Troia cultivar), when in vitro incubated with peripheral blood lymphomonocytes from patients with Ni-mediated CAD, reduced release of NO, IL-17 and IFN-γ, whereas enhancing IL-10 production they exerted antioxidant and anti-inflammatory activities [51]. In a clinical trial, oral administration of Nero di Troia red grape polyphenols to patients with Ni-mediated CAD confirmed in vitro experiments in that they decreased serum levels of IFN-γ, IL-4, IL-17, NO, and pentraxin 3, whereas levels of IL-10 were augmented [217]. This nutraceutical regimen led to an amelioration of CAD cutaneous manifestations.

With special reference to asthma models, the flavonoid polymer oligomeric proanthocyanidins reduced airway inflammation, Th2 cytokine release and antigen presentation in a mouse model of asthma [218]. Furthermore, evidence has been provided that flavones, such as luteolin and tetramethoxyluteolin acted on mast cells, decreasing release of histamine and PGD2, which are mediators implicated in asthma pathogenesis [219,220]. The above described inhibitory mechanisms seem to depend on blockade of intracellular calcium and inhibition of NF-κB [220].

Quercetin, a flavonoid contained in red grapes as well as in onions, broccoli, and apples, reduced recruitment of eosinophils and production of IL-4 and IL-5 in the bronco-alveolar fluid from mice with experimental asthma [221,222]. Cyanidin, another anthocyanidin, was able to reduce the binding of IL-17 to the IL-17RA subunit of the IL-17 receptor in a murine model of asthma [223]. Neutralization of IL-17 activity decreased inflammation and hyper-reactivity.

Food allergy is an adverse reaction to food which is mediated by IgE upon activation of Th2 cells. Dietary isoflavones have been demonstrated to suppress costimulatory molecules (CD83 and CD80) on DCs; thus, hampering activation of Th2 cells in a murine model of peanut allergy [224]. Also, in an intestinal cell model of food allergy, quercetin was able to suppress IgE-mediated allergic inflammation [225].

Autoimmune diseases share a common pathogenic mechanism of action such as the immune attack against self-components of the body [226,227,228,229,230]. Then, several factors contribute to autoimmune disease development and, among them, genetic, epigenetic, and environmental conditions should be stressed out.

In view of their antioxidant and anti-inflammatory activities, polyphenols have been used for the treatment of autoimmune disorders [231,232].

EGCG was shown to be effective in a murine model of human Sjogren’s syndrome, attenuating the TNF-α induced damage of salivary acinar cells [233].

In an experimental model of rat autoimmune myocarditis, quercetin afforded cardioprotection, decreasing phosphorylated forms of ERK1/2 and p38 [234].

RES has been shown to be very effective in type 1 diabetes either in vitro or in vivo studies [235] via increased expression of Sirt-1 [236]. In animal studies, oral or subcutaneous administration of RES to non-obese diabetic mice, led to a decreased traffic of Th1 cells and macrophages from periphery to pancreas, thus attenuating insulitis [237]. Also in a model of streptozotocin-induced diabetes in rats, RES administration by gavage prevented islet destruction [238].

In animal models of IBD, RES administration was very effective in reducing mucosal inflammation via inhibition of malondialdehyde and increase of glutathione peroxidase activities, respectively [239,240,241,242,243,244]. Furthermore, in the above models, decrease in neutrophil infiltration and proinflammatory cytokine release and increase in number of Bifidobacteria and Lactobacilli with reduction of intestinal wall fibrosis have been observed [239,240,241,242,243,244].

RES has been experimented either in vitro or in vivo in rheumatoid arthritis. Using human fibroblast-like synoviocytes, RES mitigated NADPH oxidase activity and ROS generation, increased Sirt-1 mRNA, and inhibited release of MMPs and receptor activator of NF-κB ligand [245,246,247,248]. RES also attenuated rheumatoid arthritis, blocking p38 and JNK pathways with decrease in ROS and inflammatory markers in rat RSC-364 synovial cells [249].

In rabbit arthritis, intra-articular injection of RES dramatically reduced cartilage destruction [250]. On the other hand, in various models of experimental arthritis oral administration of RES reduced severity of disease, dampening release of proinflammatory cytokines, even including IL-17 [108,251,252].

Psoriasis is an autoimmune disease mainly characterized by hyperproliferation of keratinocytes and production of IL-23 and IL-17 with inflammatory infiltrates in the dermis [253]. In vitro studies have demonstrated that RES induced apoptosis of HaCaT keratinocytes via Sirt-1 activation [254]. Furthermore, evidence has been provided that RES inhibited proliferation of normal human keratinocytes, hampering aquaporin 3 activation [255]. In a murine model of psoriasis-like skin inflammation RES attenuated skin damage, decreasing mRNA expression of IL-17 and IL-19 [256].

As far as clinical trials are concerned, patients affected by multiple sclerosis were administered with 600 mg/day of EGCG for 12 weeks [257]. At rest, metabolic responses were determined in treated patients in comparison to those administered with placebo. Results demonstrated that expenditure of post-prandial energy, glucose oxidation, and supply as well as adipose tissue perfusion were reduced in men but remained more elevated in women. During exercise, post-prandial energy expenditure was reduced in the EGCG group when compared to placebo.

Quercetin has been found to be beneficial in sarcoidosis patients, decreasing oxidative and inflammatory markers (TNF-α and IL-8), when administered at a dose of 4 × 500 mg within 24 h [258].

In a double-blind trial supplementation of RES to UC patients (500 mg/day for six weeks) reduced clinical manifestations, decreasing oxidative stress. [259].

The effects exerted by polyphenols on allergy and autoimmune diseases are synthesized in Table 7.

## 5. Discussion

The effects of polyphenols either as a dietary source or as supplements have intensively been investigated. Molecular studies have revealed the activity of these compounds on major signaling pathways. Moreover, different cell receptors for polyphenol binding have been characterized, thus indicating their capacity to modulate endocrine, metabolic and immune functions.

Among several activities they may exert, polyphenols are endowed with antioxidant and anti-inflammatory functions which justify their employment in different human diseases, as discussed in the present review. Nevertheless, there is still a lack of knowledge about the exact polyphenol concentration in foods and drinks, their degree of absorption as well as metabolism in human body. Another issue to be clarified is the assessment of which compound accounts for a given function, since a plethora of polyphenols are absorbed via dietary source. It seems that a combination of polyphenols rather than a single compound may lead to more effective beneficial effects.

Quite importantly, evidence has been provided on the effects of grape and red wine polyphenols on gut microbiota [260]. On the other hand, gut microbiota may account for the formation of a number of polyphenolic metabolites that may contribute to human health effects. However, due to the individual variations in microbiota composition, more studies are needed for a better understanding of the mutual interaction between polyphenols and gut microbiota.

Finally, one should take into consideration that polyphenols, when used as nutraceuticals and/or cosmetics, raise problems of safety and toxicity in view of their increased bioavailability and biological activity. In fact, some dietary supplements contain concentrations of polyphenols 100 times more elevated than those related to a western diet [261]. In a number of studies, administration of antioxidants has caused severe side effects such as mortality or stroke [262,263,264,265]. In this context, the possible interaction between polyphenols and drugs requires more intensive studies to understand the existence of synergism or neutralization in relation to their therapeutic activity.

## Figures and Tables

**Table 1 antioxidants-09-00035-t001:** Red grape polyphenol-induced antioxidant and anti-inflammatory activities.

Polyphenol	Activity
Quercetin	Inhibition of: COX, PPARγ, eNOS, in rodent macrophages [47,48,49]
Quercetin, epigallocatechin-gallate	Inhibition of: NF-κB translocation and phosphorylation of IκBα proteins in macrophages and microglia [53,54,55,56,57,59]; MAPK pathway with reduced release of TNF-α and IL-12 in immune and non-immune cells [63,64]
Quercetin, epigallocatechin-gallate, red wine	Inhibition of arachidonic acid pathway via reduction of prostaglandin and leukotriene release, inhibiting PLA2, COX and LOX [67]

Abbreviations: COX: cyclo-oxygenase, eNOS: endothelial nitric oxide synthase, IL: interleukin, LOX: lipoxygenase, MAPK: Mitogen-activated protein kinases, NF-κB: Nuclear factor kappa-light-chain-enhancer of activated B cells, PLA2: Phospholipase A2, PPAR: peroxisome proliferator activated receptor, TNF: tumor necrosis factor.

**Table 2 antioxidants-09-00035-t002:** Red grape polyphenol-induced immunomodulation.

Polyphenol	Activity
Quercetin, red wine-derived polyphenols	Inhibition of DC and monocyte function with reduced production of proinflammatory cytokines and chemokines [85,86]
Fisetin	Inhibition of Th1 and Th2-related cytokines in vitro [87]; Suppression of murine delayed-type hypersensitivity in vivo [89];
RES	Activation of Sirt-1 with disruption of the TLR-4/NF-κB/STAT pathway and decreased production of cytokines, PAF and histamine [90,91,92]; Induction of AMP-activated protein kinase with increased levels of NAD+ which, in turn, activates Sirt-1 [97]; Inhibition of the NLRP3 inflammasome [103]; Inhibition of the GM-CSF, IL-1β and IL-6 in the context of atheroma [104,105,106,107]; Inhibition of IL-17 release by Th17 cells and increase of IL-10 by Treg cells [108,109,110]; Increase of NK cell activity against leukemia and lymphoma cells via up-regulation of perforin expression and decrease of bacterial burden and mortality in acute pneumonia in rats [113,114,115]

Abbreviations: DC: dendritic cell, GM-CSF: granulocyte-macrophage colony stimulating factor, IL: interleukin, MAPK: mitogen-activated protein kinases, NAD: nicotinamide adenine dinucleotide, NF-κB: nuclear factor kappa-light-chain-enhancer of activated B cells, NK: natural killer, NLRP3: NLR pyrin domain containing 3, PAF: platelet activating factor, ROS: reactive oxygen species, Sirt-1: sirtuin-1, STAT: signal transducer and activator of transcription, Th: T helper, TLR: Toll-like receptor, TNF: tumor necrosis factor, Treg: T regulatory cells.

**Table 3 antioxidants-09-00035-t003:** Effects of red grape polyphenols on obesity/diabetes.

Polyphenols	Disease	Activity
Gallic acid	Obesity	Reduction of body weight in rodents with inhibition of lipid droplet formation in the liver or adipose tissue, and normalization of lipid profile [128,129,130,131]
Red grape polyphenols from Nero di Troia red grape cultivar	Obesity	In vitro experiments demonstrated inhibition of IL-21/IL-17, IL-1β and TNF-α release from obese lymphomonocytes with increase of IL-10 [132]
Quercetin, epicatechins	Diabetes	Protection of β cell survival with inhibition of NF-κB activation and ROS generation [139]

Abbreviations: IL: interleukin, NF-κB: nuclear factor kappa-light-chain-enhancer of activated B cells, ROS: reactive oxygen species, TNF: tumor necrosis factor.

**Table 4 antioxidants-09-00035-t004:** Effects of red grape polyphenols on inflammatory bowel disease.

Polyphenols	Disease	Activity
Fermented grape marc	DSS-induced murine colitis	Abrogation of intestine length shortening [148]; Decreased content of inflammatory cytokines in intestinal homogenates [148]
Bronze tomatoes red grape skin	DSS-induced murine colitis	Improvement of: stool consistency, fecal blood content and weight loss [149]
RES	Rat-induced colitis (2,4,6-trinitrobenzene sulfonic acid model)	Reduction of: PG, COX-2 expression, neutrophil recruitment and TNF-α release [150]
RES	DSS-induced murine colitis/UC	Decrease of: IL-6 expression, apoptosis, mitochondrion fatty acid oxidation, Wnt signaling, iNOS expression and NF-κB activation in murine colitis; Up-regulation of Treg cells and amelioration of clinical symptoms [151,152]
RES	IL-10^−/−^ mouse model of IBD	Activation of myeloid derived suppressor cells and reduction of inflammation [153,154]

Abbreviations: COX-2: ciclo-oxygenase-2, DSS: dextran sulfate sodium, IBD: inflammatory bowel disease, IL: interleukin, iNOS: inducible nitric oxide synthase, NF-κB: nuclear factor kappa-light-chain-enhancer of activated B cells, PG: prostaglandin, RES: resveratrol, TNF: tumor necrosis factor, Treg: T regulatory cells, UC: ulcerative colitis.

**Table 5 antioxidants-09-00035-t005:** Effects of red grape polyphenols on neurodegeneration.

Polyphenols	Disease	Activity
Red grape skin and GSSE	Murine PD	Protection of neurons against 6-OHDA-induced damage with decrease in apoptosis, ROS production and inflammatory markers [176]
Quercetin	Murine AD	Inhibition of TLR-4 signaling and reduced expression of TLR-4 and TLR-2 [178,179]
RES	LPS and Aβ-mediated microglia neuroinflammation	Inhibition of TLR-4/NF-κB/STAT pathway [180]
EGCG	LPS-impaired adult hippocampal neurogenesis	Inhibition of TLR-4 [181]
RES	AD (clinical trial)	Decrease in neuro-inflammation and in liquoral levels of Aβ40 and increase in dentate-gyrus-related cognitive functions and hippocampal memory [184,185]
EGCG	MSA (clinical trial)	No effects [190]

Abbreviations: Aβ: Amyloid β, AD: Alzheimer’s disease, EGCG: epigallocatechin gallate, GSSE: grape seed and skin extract, 6-OHDA: 6-Hydroxydopamamine, IBD: inflammatory bowel disease, LPS: lipopolysaccharide, MSA: multiple system atrophy, NF-κB: nuclear factor kappa-light-chain-enhancer of activated B cells, PD: Parkinson’s disease, RES: resveratrol, ROS: reactive oxygen species, STAT: signal transducer and activator of transcription, TLR: Toll-like receptor.

**Table 6 antioxidants-09-00035-t006:** Red grape polyphenol effects on cancer.

Polyphenols	Effector Cells	Activity
RES	Treg cells	Decrease in Treg cell frequency in murine renal carcinoma, and Eg-7 (syngenic lymphoma) with reduced release of TGF-β and increased production of IFN-γ by intranodal CD8+ cells [197]
EGCG	Human chronic lymphocytic leukemia (clinical trial)	Decrease of Treg cells and serum levels of IL-10 and TGF-β [198]
RES	TAM cells (murine cancer)	Suppression of STAT3, inhibition of lymphangiogenesis and deactivation of M2 macrophages [203]
RWE	Murine cancer	Suppression of angiogenesis and induction of apoptosis, reduction of precancerous lesions [208,209,210]
Muscadine grape skin extract	Prostate cancer	Induction of autophagy with apoptosis of cancer cells [211]
Liofenol^TM^ (RWE)	Colon cancer cells	Arrest of cell growth with increase in p53 and p21 protein expression [212]

Abbreviations: EGCG: epigallocatechin gallate, IFN: interferon, IL: interleukin, RES: resveratrol, RWE: red wine extracts, STAT: signal transducer and activator of transcription, TAM: tumor associated macrophages, TGF: transforming growth factors, Treg: T regulatory cells.

**Table 7 antioxidants-09-00035-t007:** Effects of grape polyphenols on allergy and autoimmune diseases.

Polyphenols	Effector Cells/Disease	Activity
FGM	Rat basophilic leukemia cells	Inhibition of IgE binding to cells [215,216]
Polyphenols extracted from seeds of red grape (Nero di Troia cultivar)	Peripheral blood lymphomonocytes from Ni-mediated CAD	In vitro decrease of: NO, IL-17 and IFN-γ release with increase of IL-10 release [51]
Polyphenols extracted from seeds of red grape (Nero di Troia cultivar)	Ni-mediated CAD	In vivo decrease of: serum levels of IFN-γ, IL-4, IL-17, NO and pentraxin 3 with increase of serum IL-10 [217]
Flavones	Murine asthma mast cells	Decrease of histamine and PGD2 [219,220]
Quercetin	Murine asthma	Reduction of eosinophil recruitment and IL-4 and IL-5 levels in bronchoalveolar fluid [221,222]
Cyanidin	Murine asthma	Decrease of IL-17 binding to the IL-17RA subunit of the IL-17 receptor [223]
Isoflavones	Murine model of peanut allergy	Suppression of costimulatory molecules (CD83 and CD80) on DCs with reduced activation of Th2 cells [224]
Quercetin	Food allergy	Suppression of IgE-mediated allergic intestinal inflammation [225]
EGCG	Murine Sjogren’s syndrome	Decrease in TNF-α-induced damage of salivary acinar cells [233]
RES	Rat RSC-364 synovial cells	Blockade of p38 and JNK pathways and decrease of ROS and inflammatory markers [249]
Quercetin	Rat autoimmune myocarditis	Cardioprotection via decrease of phosphorylated ERK1/2 and p38 [234]
RES	T1D	-Decrease of in vitro apoptosis via increased Sirt-1 expression [236]; In vivo, in an obese model attenuation of insulitis due to diminished traffic of Th1 cells and macrophages from periphery to pancreas and prevention of islet destruction [237]
RES	IBD	Reduction of mucosal inflammation via decrease of: malondialdehyde, COX-2, PGE-synthase 1, TGF-β, neutrophil infiltration and increase of: glutathione peroxidase activity, Bifidobacteria and Lactobacilli [239,240,241,242,243,244]
RES	Rheumatoid arthritis	In vitro, using, fibroblast-like synoviocytes, decrease in: NADPH oxidase activity, MMP release, RANKL and ROS generation with increase in Sirt-1 mRNA [245,246,247,248]; In experimental models, reduction of IL-17 and reduction of cartilage destruction [250]
RES	Psoriasis	In vitro induction of keratinocyte apoptosis via Sirt-1 activation and keratinocyte inhibition via decrease of aquaporin 3 activation [254,255]; In an in vivo model of murine psoriasis decrease in mRNA expression of IL-17 and IL-19, thus, mitigating skin damage [256]

Abbreviations: CAD: contact allergic dermatitis, COX-2: cyclo-oxygenase-2, DCs: dendritic cells, EGCG: epigallocatechin gallate, ERK: extracellular signal-related kinases, FGM: fermented grape marc, IBD: inflammatory bowel disease, IFN: interferon, IL: interleukin, JNK: c-Jun amino-terminal kinases, MMP: metalloproteinases, NADPH: nitrate reductase, Ni: nickel, NO: nitric oxide, PG: prostaglandin, RANKL: receptor activator of nuclear factor kappa-Β ligand, RES: resveratrol, ROS: reactive oxygen species, Sirt-1: sirtuin-1, T1D: type 1 diabetes, TGF: transforming growth factors, Th: T helper cells, TNF: tumor necrosis factor, Treg: T regulatory cells.

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
