# Peer review of "Recent Advances on the Anti-Inflammatory and Antioxidant Properties of Red Grape Polyphenols: In Vitro and In Vivo Studies"

_antioxidants, 2019, doi:10.3390/antiox9010035_

Round 1
Reviewer 1 Report
Manuscript is very interesting and it provides detailed knowledge on health benefits of red grape phenols, both in vitro and in vivo. Concept of manuscript is very well established and authors took good effort to summaries studies related to this topic. It is good to have everything in one place about red grape phenols. Some technical corrections should be done in order to accept manuscript.
Correct tables, put them in one page, and correct technical mistakes in it like unnecessary space
Be careful that you list all abbreviations, some are missing. Also correct abbreviations: COX, cyclo-oxygenase – it would be better to write it in this way COX - cyclo-oxygenase
Line 440 – unbold words
Reference list
References 1 – 3 – ISBN is not written correctly; pages are missing
Author Response
Please, see the attachment.

Reviewer 2 Report
Overview:
In this manuscript, the authors review the current state of the field in regards to the anti-inflammatory and anti-oxidant effects of red grape polyphenols and their effects in many diseases.
Comments:
1. The authors take great care to make this review quite comprehensive covering a large number of studies and integrating a wide array of concepts related to polyphenols and their health effects. However, in this pursuit for broad inclusion, many important concepts are not introduced fully and are discussed with language that is somewhat unclear and imprecise.
This is particularly true in the introduction. The manuscript would greatly benefit from a large-scale edit of this section with greater attention paid to the structure of the text. Perhaps breaking up the first paragraph into multiple paragraphs with single or few specific points would streamline the text and better orient the reader as the direction the authors are going. Currently, there are so many different concepts jammed into this one paragraph the reader can’t help but feel overwhelmed and confused.
The latter sections (particularly Section 4) discussing specific studies and diseases are much more specific and the statements crisper and easier to comprehend, understand and appreciate.
2. As stated above, the authors make a wonderful attempt to make this review as broad and comprehensive as possible, yet this reviewer was left with an overwhelming sense that only one side of the evidence was being presented. For instance, in Section 2 the authors repeatedly discuss the anti-inflammatory and anti-oxidant effects of polyphenols and attribute this mainly to inhibition of the key inflammatory actors. However, many of these enzymes and proteins and their regulation is quite complex.
One specific example is in the authors’ discussion of the metabolism of arachidonic acid. The authors state, “Among other mechanisms of anti-inflammation promoted by polyphenols, inhibition of arachidonic acid (AA) pathway plays a paramount role.”
While certainly AA can be metabolized into pro-inflammatory mediators like leukotrienes and prostaglandins, LOX also converts AA into anti-inflammatory (or pro-resolving) compounds (e.g. Lipoxins). Further, signaling of the prostanoids (particularly PGE2) is much more intricate than simply being lumped into a “pro-inflammatory” bin. In fact, PGE2 has been shown to be a critical initiator of the resolution of inflammation program.
Discussing inflammation in such black and white terms there is a large amount of nuance that is being ignored. This reviewer feels that this lack of balance in discussing the actions of polyphenols is missing throughout the manuscript.
3. The authors do a nice job of presenting examples of confusion or lack of clarity in the field yet then make no attempt to rectify the disparity. For instance, in Section 4.1 the authors point out the clinical trials with gallic acid have resulted in opposite results, but they don’t mention what may account for these differences. Was it due to differences in the study design or population (obese vs. pre-obese individuals)?
4. Also in Section 4.1, the authors very thoroughly and specifically describe so much of what has been shown in obesity, but then only mention diabetes with minimal details. This reviewer would have liked to see this subsection expanded, especially in regards to the protection of pancreatic B-cells, as the cited studies appear quite impactful.
Author Response
Please, see the attachment.

Reviewer 3 Report
This review paper presented by Magrone et al. gathers the most relevant and updated research carried out with polyphenols from red grapes, as antioxidant and anti-inflammatory compounds, as well as on selected major diseases (obesity, cancer, among others). Although it can be very interesting for the readers, and tables are helpful summarizing items, it would be positive for the manuscript to provide a more structured text. For example, headings 2 and 3 would be benefitted from adding subheadings to avoid such a condensed, and somehow confusing, text (i.e. it could be structured by pathways modulated by polyphenols). Moreover, inclusion of new paragraphs in heading 2 would be acknowledged.
Other minor comments:
Lines 131 and 211: "Table 1" and "Table 2" are in bold style, however, the rest of Tables is in normal style. Lines 68 and 220: Remove "For space limitations". Line 440: "receptor activator of NF-kB ligand" is in bold style. Please, fix. Avoid repetitive use of words.
English style should be addressed, text should be clearer presented and better structured
Author Response
Please, see the attachment.

Reviewer 4 Report
This review covers a wide range of health benefits of polyphenols, and may give some impacts to researches in the related fields. One thing I would like to request is that the authers would explain more clearly the reason why they chose red grape polyphenols in relation to their chemical composition.
Author Response
Please, see the attachment.
